# Examining the Relationship between Cost and Quality of Care in the Neonatal Intensive Care Unit and Beyond

**DOI:** 10.3390/children7110238

**Published:** 2020-11-19

**Authors:** Lauren Culbertson, Dmitry Dukhovny, Wannasiri Lapcharoensap

**Affiliations:** Department of Pediatrics, Oregon Health and Science University, Portland, OR 97239, USA; culbertl@ohsu.edu (L.C.); dukhovny@ohsu.edu (D.D.)

**Keywords:** health care costs, quality, neonatal intensive care, value

## Abstract

There is tremendous variation in costs of delivering health care, whether by country, hospital, or patient. However, the questions remain: what costs are reasonable? How does spending affect patient outcomes? We look to explore the relationship between cost and quality of care in adult, pediatric and neonatal literature. Health care stewardship initiatives attempt to address the issue of lowering costs while maintaining the same quality of care; but how do we define and deliver high value care to our patients? Ultimately, these questions remain challenging to tackle due to the heterogeneous definitions of cost and quality. Further standardization of these terms, as well as studying the variations of both costs and quality, may benefit future research on value in health care.

## 1. Introduction

The Institute for Healthcare Improvement (IHI) launched the term Triple Aim in health care [1] to encourage institutions to focus on improving the experience of care, improving the health of populations, and reducing per capita costs of health care. Inherent within Triple Aim is the understanding that more spending does not necessarily equate improved experience and population health.

From 1996 to 2013, total health care spending on children increased from USD 149.6 billion to USD 233.5 billion (2015 U.S. dollars). In 2013, the largest health condition leading to health care spending for children was well-newborn care in the inpatient setting [2], yet the United States (U.S.) ranks last among eleven high-income countries in infant mortality [3], highlighting the paradox of “spending more and achieving less” [4].

Through this article, we explore what is known about the relationship between costs and outcomes in the adult and pediatric literature. We will then review the costs associated with neonatal intensive care and the current efforts toward reducing unnecessary spending. We will discuss value in health care and the complex interplay between value in the context of costs and outcomes.

## 2. Health Care Spending

The Commonwealth Fund [3], a private U.S. foundation that supports independent research on health care issues to promote high-performing health care systems with improved quality and efficiency, develops a report every three to four years that compares the health care system performance of the U.S. and ten other, similar, high-income countries. The report is developed using data from Commonwealth Fund International Health Policy Surveys, reports of the Organization for Economic Cooperation and Development, the European Observatory on Health Systems and Policies, and the World Health Organization. Seventy-two measures relevant to health care systems performance were identified and organized into five main performance domains (care process, access, administrative efficiency, equity, and health care outcomes). The report summarizes quality of care in these main aforementioned performance domains and the costs as the per capita spending. According to the Commonwealth Fund *Mirror*, *Mirror* 2017 report [3], the U.S. ranks last in the overall ranking and either last or second to last in four of the five domains. A specific example cited within health care outcomes domain is infant mortality. According to their raw data, the U.S. has 6.0 deaths per 1000 live births, performing well below even the next lowest performing country (Canada at 4.8 per 1000 live births) and a stark contrast to the top-ranking country, Sweden, with 2.2 deaths per 1000 live births. Although there is variability among each of these health care systems, despite performing poorer, as the report highlights, the U.S. spends more money on health care than any of the other included ten counties, and this amount continues to rise at a steeper slope.

## 3. Stewardship Campaigns: Reducing Waste

One method of reducing costs is with health care stewardship campaigns such as *Choosing Wisely*, where specialties have created lists of tests, treatments and procedures within their respective fields to avoid because they are considered low value (i.e., unnecessary and/or increase costs without improving outcomes) [5,6]. Since its beginning in 2012, *Choosing Wisely* has expanded from the “top five” lists of nine specialty societies to more than seventy societies in 2020. A variety of initiatives have focused on this list with promising improvements such as a reduction in repetitive laboratory testing which not only increases costs, but also can lead to poor patient outcomes [7]. Other interventions to reduce costs and waste capitalize upon the power of electronic health records (EHR), such as including price display in computerized physician order entry or *incorporating Choosing Wisely* recommendations into clinical decision support tools in the EHR. A systematic review of 19 studies by Silvestri et al. [8] found that price display interventions for laboratory tests, imaging studies, or medications likely reduces order costs to a modest degree; however, there was limited evidence related to patient safety outcomes, although it appeared unchanged. Utilization of clinical decision support tools have been found to impact physician behavior and reduce costs, improve length of stay (LOS), and reduce complication rates [9]. These findings suggest that reducing low-value interventions does not harm patient safety outcomes and may even improve them.

In joining with other specialties, Ho et al. [10] utilized national surveys and an expert panel through the American Academy of Pediatrics Section on Neonatal Perinatal Medicine to develop the “*Choosing Wisely* Top Five” list for newborns consisting of “(1) avoid routine use of antireflux medications for treatment of gastroesophageal reflux disease or apnea and desaturation in preterm infants, (2) avoid routine continuation of antibiotic therapy beyond 48 h for initially asymptomatic infants without evidence of bacterial infection, (3) avoid routine use of pneumograms for pre-discharge assessment of ongoing and/or prolonged apnea of prematurity, (4) avoid routine daily chest radiographs without an indication for intubated infants, and (5) avoid routine screening term-equivalent or discharge brain MRIs in preterm infants [10]”. This set of interventions and procedures adds to the list of other pediatric *Choosing Wisely* topics, has helped to escalate and continue the conversation of being health care stewards within Neonatology, and has led to large initiatives, such as *Choosing* Antibiotics *Wisely* by the Vermont Oxford Network (VON)—a national quality improvement initiative focused on reduction in unnecessary antibiotic exposure in newborns [11,12]. Amongst the next steps for health care stewardship within Neonatology is to figure out how to utilize neonatal care “more wisely” [13] to help combat the increasing trends in neonatal intensive care unit (NICU) admissions over time [14].

## 4. Defining Cost and Quality of Care

While interventions to reduce waste and cost focuses first on “do no harm”, others may hypothesize that reducing costs may inevitably lead to decreased quality of care. Some describe a framework of comparative costs and quality to define high and low performers (Figure 1).

A systematic review by Hussey et al. in 2013 [15] looked at 61 studies published between 1990 and 2012 to review the evidence of the association between health care quality and cost in the adult literature. However, the review is limited due to the variability in the type of cost and quality measures reported by the individual studies. Cost measure type included accounting costs, care intensity index, charges, and expenditures; while quality measure type included access, composite measure, outcome, patient experience, process, and structure. Associations between quality and spending were categorized as “positive”, “mixed-positive”, “negative”, “mixed-negative”, “mixed”, “no difference”, and “imprecise or indeterminate”. Overall, there was heterogeneity in the results without a clear direction or magnitude of association between measures of cost and quality. The authors concluded that the association between health care costs and quality remains poorly understood. This challenge of comparing the body of literature repeatedly occurs as there is a wide range of definitions for cost and quality (Table 1). For example, some studies use length of stay as a marker of cost of care while others use it as a measure of quality. In addition, costs are also reported as adjustments to a certain year to account for inflation.

Looking at the link between costs and quality of care of a specific example within adult medicine, there has been a substantial amount of work focused on adult acute myocardial infarctions (AMI). Globally, there have been contradictory data on whether costs and quality of care for AMI are related. Nuti et al. [16] completed a cross-sectional study looking at outcomes among Medicare beneficiaries ≥65 years hospitalized with AMI and concluded that increased costs in the in-hospital setting were not associated with differences in mortality or reduced cost of post-acute care. They also did not see improved outcomes related to increased costs, suggesting that higher cost hospitals could reduce spending and resource utilization without reducing quality of care [16]. A large study compiling data from Finland, France, Germany, Spain, and Sweden found that, specifically for AMI patients in Sweden, the costs were higher in hospitals with the highest quality of care [17]. However, there was not a clear cost–quality tradeoff when comparing cost and in-hospital mortality in the other countries. Further, a study using data from the Korean National Health Insurance Program also found no significant trade-off between cost and quality as related to AMI [18]. Of note, a significant limitation of the above studies in adult AMI data focused on mortality, whether in-hospital or 30-day post-discharge as a measure of quality of care.

## 5. Cost of Pediatric Care

The lack of evidence for a clear correlation between cost and quality is also seen in the pediatric literature. Using the Pediatric Health Information Systems (PHIS) database, which includes data compiled from 47 participating children’s hospitals across the U.S., Gupta et al. [19] evaluated the relationship between hospital costs and patient outcomes in pediatric critical care. They used in-hospital mortality as their primary quality metric with a secondary analysis using prolonged hospital length of stay (>75th percentile) as a quality metric. Using hospital costs rather than hospital charges, they compared quality outcomes among high-performance hospitals (low costs, low mortality) and low performance hospitals (high costs and high mortality) and did not find any relationship between costs and patient outcomes in children with critical illness. They found that even among patients with the same condition, hospital costs vary widely across the included children’s hospitals and the highest costs of care were associated with utilization of high-cost resources such as extracorporeal membrane oxygenation (ECMO) and inhaled nitric oxide [19].

A study using combined data from the Society of Thoracic Surgeons Congenital Heart Surgery (STS-CHS) and the PHIS database looked at the relationship between quality and cost in 27 hospitals among children undergoing congenital heart surgery between 2004 and 2010. The authors divided the hospital costs by tertiles and found that most hospitals in the lowest adjusted cost tertile delivered the highest quality of care, defined as the lowest rate of inpatient mortality, and also reported out the shortest LOS and fewer major complications [20]. However, a study from the Kids’ Inpatient Database (KID) came to the opposite conclusion—using a hospital charge-to-cost approach, authors found that higher cost hospitals had decreased mortality [21]. Despite studying similar populations (children undergoing congenital heart disease surgery), the two studies came to opposite conclusions, highlighting the difficulties in studying this very important topic due to the heterogeneity of approaches.

In efforts to find methods to decrease resource utilization without affecting patient outcomes, Lion et al. [22] studied the implementation of evidence-based, standardized clinical pathways at Seattle Children’s Hospital. They studied their fifteen clinical pathways in aggregate, tracking hospital charge-to-costs and length of stay, using patient physical functioning scores and readmission rates as balancing measures. After implementation of their pathways, they had a significant halt of rising costs with a post pathway slope difference of USD 155 per patient per month [95% confidence interval (CI)−USD 246 to −USD 64] and significantly decreased length of stay with a post pathway slope difference −0.03 days per admission per month (95% CI −0.05 to −0.02). This study is a powerful reminder of using quality improvement and standardization of care to improve patient outcomes, all while reducing costs and utilization.

## 6. Cost of Neonatal Care

The costs of health care in Neonatology are known to be large, especially for very low birth weight (VLBW, birthweight < 1500 g) infants and those with significant comorbidities of prematurity. In addition, term infants admitted to the NICU are inherently sicker than their uncomplicated counterparts in the mother–baby units and thus appropriately require a longer length of stay and incur more costs [23]. According to a cross-sectional study by Russel et al. [24], of the 4.6 million infant hospital stays in the U.S. in 2001, 8% had a diagnosis of preterm birth/low birth weight, yet the costs for this group represented 47% of the costs for all infant hospitalizations, which equates to USD 5.8 billion dollars. In contrast, 10% of the costs were for uncomplicated newborns, which comprised of 42% of total hospitalizations. Of these preterm/low birth weight infants, 25% had one of more complications (respiratory distress syndrome, bronchopulmonary dysplasia, intraventricular hemorrhage, or necrotizing enterocolitis), with total estimated costs of USD 3.1 billion for these conditions. Johnson et al. [25] estimated increased costs from development of morbidities in VLBW infants as USD 12,048 (in 2009 US dollars) for brain injury (including intraventricular hemorrhage, periventricular leukomalacia, acquired hydrocephalus), USD 15,440 for necrotizing enterocolitis (NEC), USD 10,055 for late onset sepsis, and USD 31,565 for bronchopulmonary dysplasia (BPD), controlling for sociodemographic characteristics and the presence of other morbidities.

Similar to adult and pediatric medicine, Neonatology too has demonstrated variation in costs across centers without a link to an improved outcome. Massaro et al. [26] conducted a study utilizing linked data between the Children’s Hospitals Neonatal Database (CHND) and PHIS to quantify inter-center cost variation for perinatal hypoxic ischemic encephalopathy (HIE) treated with therapeutic hypothermia. These data were reviewed in the context of favorable (defined as survival with normal magnetic resonance imaging) or adverse (death or need for gastric tube feedings at discharge) in order to examine the relationship between costs and outcomes. They found marked inter-center cost variation associated with treating HIE. Interestingly, they found that although the widest cost variation across centers was electroencephalogram (EEG) use (10-fold cost variability between centers), hospitals with low cost and favorable outcomes (highest value) ranked higher in regard to EEG costs. They also found that centers with the highest frequencies of adverse outcomes had lower relative costs [26]. By delving deeper into where the differences in spending were between centers, the study was able to tease out potential drivers for cost and how that might impact outcomes. As discussed by Clark and Spitzer in an accompanying editorial, these points highlight the importance of understanding what drives the differences in costs between centers and using this information to improve outcomes and decrease costs. As we begin to understand more of what drives cost variation, we can move towards improving value.

In addition to inter-center cost variation within neonatal care, variation for outcomes has been well established and persists even with overall improvement in quality of care. Both California Perinatal Quality Care Collaborative (CPQCC) [27] and VON [28] have demonstrated a reduction in major morbidities in VLBW infants over time, yet a variation between centers persists. CPQCC noted a nearly 20% center variation over time (2008–2017). The exact reasons for such wide variation in outcomes are multi-factorial and not fully understood. Practice patterns, such as 40-fold antibiotic utilization across NICUs without a difference in positive blood stream infections [29], may help address some of it. Racial/ethnic disparities in perinatal health, which result in part from structural racism and social determinants of health such as poverty, food insecurity, socioeconomic status, and environmental toxicity (i.e., pollution), certainly contribute to both inter and intra-NICU variability. The assessment of costs and outcomes together is common practice for economic evaluations such as cost effectiveness, cost benefit and cost consequence analyses. There are many examples in Neonatology literature that evaluate the financial impact of a new therapy or program in order to globally assess what is the incremental gain in outcome for the incremental dollar amount spent, including the use of antenatal corticosteroids in preterm [30] and late preterm infants [31], the use of surfactant in respiratory distress syndrome [32], the use of exclusive human milk and human milk fortifiers in prevention of NEC in VLBWs [33,34,35,36,37], the use of probiotics in prevention of NEC in VLBWs [38], routine screening for retinopathy of prematurity [39,40], among others. These assessments are important to understand the financial impact of tests, treatments and programs. Yet similar to quality improvement and outcomes assessment at the center level, it is also important to look at costs and outcomes together in order to help understand the variation that exists and identify opportunities to reduce unnecessary spending without affecting the outcomes.

## 7. How Do We Evaluate Value in Neonatal Care?

As we continue to strive to reduce costs in medical care through quality improvement efforts or initiatives such as *Choosing Wisely*, it is important to measure the impact that these changes have on patient outcomes and strive for a focus on high-value care. In the literature, value is defined as “health outcomes achieved per dollar spent” [41]. Although value can be difficult to study well in medicine, there are multiple discussions and approaches to value in neonatal care in the literature [41,42,43,44,45].

One approach attempts to simplify the interrelationship of value, quality improvement, evidence-based medicine, and evidence-based economics with a “value equation”. In the neonatal value equation, Dukhovny et al. describe that value is determined by the outcomes (the numerator) divided by cost (the denominator) (Figure 2) [42]. The outcomes represent the interplay between quality, efficacy and safety while costs represent resource tallies as well as dollars. Ho et al. [43] build on this and describe the importance of using value and cost measures within the SMART aim (specific, measurable, attainable, relevant, time-bound) for quality improvement initiatives. With this equation, an increase in value can also be derived from improving outcomes without increasing resource utilization.

Another formula proposed by Kaempf et al. [44] calculates value in their “benefit and value metric”, which assesses quality as a composite of major neonatal morbidities and mortality as tracked by VON. Kaempf uses length of stay (LOS) as a proxy for costs, which is reasonable in the NICU population given the discussion by both Kaempf [44] and Ho et al. [43] as it represents costs and resource utilization incurred over time. Similarly, in this metric, efforts to improve morbidities and mortality would increase the value of care; however, it does not specifically address how value might be increased by decreasing LOS.

When value is studied in medicine, we are attempting to answer the question: Is it worth the investment? As addressed by Flanagan [46], the value calculations in pediatric (and neonatal) care vary from the adult value calculations in that the pediatric value calculations require a longer time course as “adult health depends on child health”. Hospitalization costs, whether for birth or within the first year of life, have discrete timepoints, but do not encompass the entirety of costs that extend beyond what can be truly reasonably studied. Even a single disease can have significant long-term impacts to the family and incur continuous costs that cannot be accounted [47]. Kaempf et al. [44] also highlight the importance of the interplay between all points of the value equation, if there is an increase in the “costs” (LOS in this case), yet it results in a significantly improved outcome (lower morbidities among VLBW infants), our overall value has improved despite the increased LOS.

While these formulas neatly summarize and conceptualize that there is a close relationship and interplay between value, cost, quality and outcomes, the question of value of care can be so complex and nuanced to study since either part of the equation: costs (whether in dollars or LOS) and quality (what defines quality and what is the right outcome to look at) each have their own challenges. Quality and outcomes can have different definitions depending on whether it is from the perspective of a health care provider, health care institution, insurance company, family, patient, or society. For example, LOS, as used in the Kaempf metric, is often simplified to be a marker for cost. However, it does not account for the significant non-medical costs to the family, fixed costs of care in the NICU, including space and staffing, as well as expected LOS for the physiologic maturation [42,43,48].

There are other considerations when trying to study value. First is the impact on family—with the potential discordance between the value from the hospital perspective and the value from the family perspective [42]. Occasionally, there exists a trade-off between family costs and health care cost. Second, there is the difficulty of proper attribution of outcomes and costs to specific hospitals, should the infant be transferred. Third, attempts to understand value from a health disparity lens are extremely complicated. One of the five domains evaluated by the Commonwealth Fund is equity and, to our knowledge, none of the cost-effectiveness and value literature specifically addresses disparities in neonatal care. Furthermore, the focus on the link between cost and quality is often focused on inpatient medicine, in particular for newborns. Some of that is inherent to the U.S. health care system that has multiple payers, and even with the introduction of the medical homes and Accountable Care Organizations, has silos of episodes of care that frequently are financially rewarded for each episode. Such an approach limits the capture of outpatient services and community-based programming that focuses on both pre-conception, pre-natal and post-natal support. Although these programs are also associated with costs, investment pre- and post-birth may have substantial improvement on both cost reduction during the inpatient stay, as well as quality of care, including closing the racial disparity gap.

Recently, there has been a growing body of literature looking at disparities in neonatal care demonstrating that there is variation in terms of clinical outcomes. Using data from the CPQCC, Profit et al. developed a validated tool (Baby-MONITOR) to assess the quality of care at different hospitals and demonstrated hospital-to-hospital variation [45]. Furthermore they compared the quality of care among different races and ethnicities between and within individual NICUs and found significant variation in care when comparing white, African American, Hispanic and Asian American populations within and between individual NICUs [49]. Their study highlights an important piece of high value care that can be overlooked when populations are evaluated as a whole. As we strive to improve value in our NICUs, we have not truly achieved this goal if such disparities persist.

## 8. Next Steps

The large variation in both costs and outcomes coupled with limited evidence in health care to demonstrate the link between spending more money (higher health care costs) and better outcomes (higher quality) presents a dilemma when it comes to investing additional resources in a system that is already constrained. In order to reduce unnecessary health care spending, we must couple cost analyses with outcome studies and evaluate the relationship between the two. While quality improvement initiatives are often used as a tool to improve outcomes, there is also the necessary investment of resources (Figure 3). Therefore, trying to study the delicate balance between what costs are important and necessary and the quality of care delivered inherently comes with many confounders. It is possible to imagine that with greater investment of resources and thus increased costs, the quality of care delivered may be improved. Conversely, higher quality of care could also lead to decreased health care costs due to lower rates of complications, morbidities, and length of stay. Furthermore, utilizing mortality as a marker for quality of care can be problematic in and of itself, as mortality could also arguably decrease cost by reducing the length of stay and reducing resource utilization (in particular in the NICU population).

There is already a model that exists in collaborative quality improvement for centers to learn from best and worst performers in order to learn opportunities for improvement. The same approach should routinely incorporate costs into the conversation in order to allow tools such as *Choosing Wisely* and EHR to drive down unnecessary spending without affecting outcomes. There are both micro and macro-economic opportunities for cost reduction while allowing additional financial investments as new technologies, drugs and programs are introduced.

## 9. Conclusions

The link between health care cost and quality of care is inconsistent across the spectrum of ages and disease processes. Much of the struggle lies in the heterogenous definitions of cost, quality, and value of care. While some studies (e.g., Kaempf Benefit Metric [44]) support a correlation between increased LOS (proxy for cost) and improved outcomes (neonatal morbidities), the literature either fails to look at the cost and quality in the same analysis or does not demonstrate a correlation. In addition, health care stewardship campaigns such as *Choosing Wisely* are built on the premise that eliminating services that are not necessary will not worsen outcomes (and potentially would improve them). Coupled with comparative reports such as *Mirror Mirror* [3], where the U.S. spends double or triple per capita on health care compared to economic equivalents yet has worse outcomes, this combination suggests that there is room to reduce costs without negatively affecting outcomes (as has been demonstrated by multiple studies). The next steps for us to improve value in health care is to better define how we look at quality and costs in order to consistently study them together. As we strive to understand the variation in clinical outcomes, we must also strive to understand the variation in costs and embrace the opportunity to reduce each rather than assume that one hospital has better outcomes because they are resource rich and allocate higher dollars. In addition, we should include a “health equity lens” to ensure that our value-based care works to eliminate disparities between groups in health care and we are truly achieving improved outcomes for all of our patients.

## Figures and Tables

**Figure 1 children-07-00238-f001:**
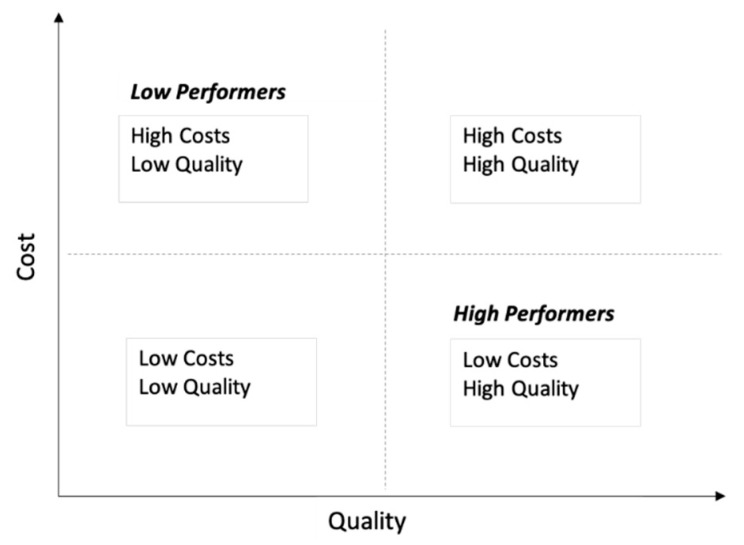
Framework used to evaluate the association between cost and quality hospitals can be classified as high or low performers based on their costs and quality of care.

**Figure 2 children-07-00238-f002:**
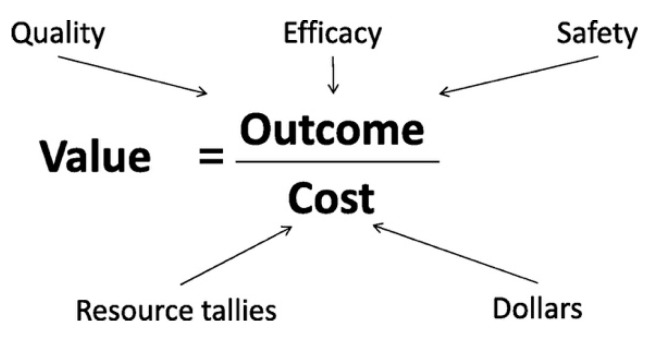
The neonatal value equation. Reproduced with permission from *Pediatrics* Vol. 137, Issue 3, e20150312. Copyright © 2016 by the American Academy of Pediatrics (AAP).

**Figure 3 children-07-00238-f003:**
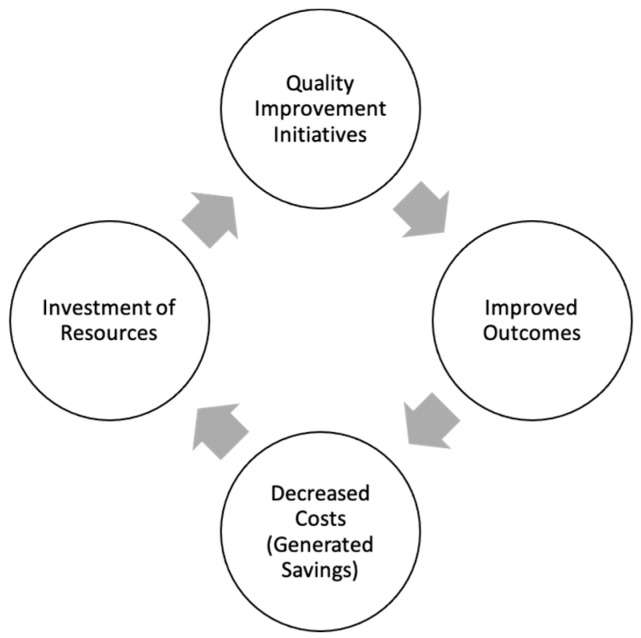
The relationship between quality improvement, outcomes, and costs.

**Table 1 children-07-00238-t001:** Examples of defining cost and quality in the literature.

Cost	Quality
Expenditure (hospital, staffing, pharmacy, etc.) Charges *: - facility and professional fees Charges * (with or without cost adjustment using Cost to Charge Ratios) Payments collected * Length of Stay Resource tallies with applied costs from literature	Morbidity Mortality Patient Safety Outcomes Complications Composite Measures Post-Acute Care Length of Stay

* Sources include hospital records, payers, all claims databases.

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
