# Peer review of "Examining the Relationship between Cost and Quality of Care in the Neonatal Intensive Care Unit and Beyond"

_children, 2020, doi:10.3390/children7110238_

Round 1

Reviewer 1 Report

This paper mainly performed literature reviews on the relationship between cost and quality of care in adult, pediatric and neonatal literature. The conclusion is that there are only limited evidence that demonstrate the link between higher health care costs and higher quality, due to the large variation in both costs and outcomes. However, to me, this conclusion is natural because the existence of so many confounders and different types of costs and definitions of outcomes. It seems the paper just listed all conclustions in previous literatures to get this conclusion. I would like the authors investigate deeper into this problem.

1) The study has not included sufficient information to readers. The paper didn't try to figure out what kinds of cost would improve the outcome, nor did the paper make a detailed comparion on the specific factors that lead to different conclusions in previous literatures.

2) There are only few quantitative measurements and analysis about studies for different literatures. More statistics will give a better comparison.

3) To show clearer comparisons among literatures, it may be better to summarize cost types, definitions of outcomes, study designs, results and conclusions into tables across different literatures. 

In general, I think this paper requires more work to get published.

Reviewer 2 Report

Line 25-26 From "1996 to 2013, health care spending on children increased from $149.6 billion to $233.526 billion - when accounting for inflation this is only a 5% increase - the way it is written leads a reader to perceive a greater difference then there is in actuality. 

It would be good if you clarified total spending versus per child spending.

Line 39 - you gloss over - care process, access, administrative efficiency, equity, and health care outcomes 

Reading Lines 44 - 48 - It should be clarified that you are aware that the healthcare system in the US and Sweden are very different - e.g., variability in per child spending.

I think a bit more critique of existing results is necessary.

line 52 - the jump into stewardship is rather abrupt and it is not clear why you chose this direction.  Are you making a case for that being a cause or the cause of spending.

LINE 166 understand more of what drives cost variation, we can move towards improving value. - this is a good statement - it would be important to see more interpretation what the literature 

The severity of the illness is not taken into account - ICU babies are going to be sicker

Part 7 is good - it just seems like it was a bit of a meandering long road to get there.

Reviewer 3 Report

This is a helpful compilation of information from the literature, on the relationships between cost and quality of medical care. It is not original work but clearly lays out various issues.  A missing factor in this relationship is the environmental and societal issues of the patient and the community.  Community supports enhance health and lead to better outcomes, e.g. lower rates of infant mortality, and are thought to be more important than actual medical care.  Society and environment contribute to fewer prenatal births and so infant mortality can be lower regardless of healthcare / neonatal nursery expenditures.  It would be appropriate for the authors to acknowledge this.

Nevertheless, the  paper lays out the importance of looking at cost and quality, and provides a scholarly summary of what is known.

Specific comments:

Lines 53-55: there is a verb missing, this is not a full sentence.

Lines 81-83: this would be one place to bring in predeterminnig factors of society and environment; as well as prenatal care in outpatient setting.

Lines 97-98: Maybe I am not reading the chart correctly, but I think High and Low performers might be reversed (i.e. High performers have low cost, high quality; but they are on the graph on high cost and low quality)

Line 109-110: I think there is a word missing in the sentence. "...patients in Sweden the costs were higher..."?
Line 153-167: not sure what your point is here.

Line 175: please refer to SES and environment. racial/ethnic disparities are due to poverty, disenfranchised communities, pollution, etc.

Round 2

Reviewer 1 Report

The author generally addresses all my concerns. I agree with you on the possible reason of the different conclusions between cost and neonatal care: higher cost may lead to good neonatal care; on the other hand, good neonatal care may lead to low cost. So that the association between cost and neonatal is too complex to get a positive or negative conclusion. It's the ratio of the neonatal care over cost that really matters. 

Author Response

Thank you for your review of our paper.

Reviewer 2 Report

no further suggestions at this time.

Author Response

Thank you for your review of our paper.

Reviewer 3 Report

I see that the authors have addressed several comments from reviewers. The paper looks good to me.

Author Response

Thank you for your review of our paper.